# Celecoxib Microparticles for Inhalation in COVID-19-Related Acute Respiratory Distress Syndrome

**DOI:** 10.3390/pharmaceutics14071392

**Published:** 2022-06-30

**Authors:** Monica-Carolina Villa-Hermosilla, Sofia Negro, Emilia Barcia, Carolina Hurtado, Consuelo Montejo, Mario Alonso, Ana Fernandez-Carballido

**Affiliations:** 1Department of Pharmaceutics and Food Technology, School of Pharmacy, Universidad Complutense de Madrid, Plaza de Ramón y Cajal s/n, 28040 Madrid, Spain; mcvhermosilla@ucm.es (M.-C.V.-H.); ebarcia@ucm.es (E.B.); marioalonsogonzalez@ucm.es (M.A.); afernand@ucm.es (A.F.-C.); 2Institute of Industrial Pharmacy, School of Pharmacy, Universidad Complutense de Madrid, Plaza de Ramón y Cajal s/n, 28040 Madrid, Spain; 3Department of Health and Pharmaceutical Sciences, School of Pharmacy, CEU San Pablo University, 28668 Madrid, Spain; carolina.hurtadomarcos@ceu.es (C.H.); montejo@ceu.es (C.M.)

**Keywords:** celecoxib, COVID-19, PLGA, microparticles, macrophages, inhalation

## Abstract

Inhalation therapy is gaining increasing attention for the delivery of drugs destined to treat respiratory disorders associated with cytokine storms, such as COVID-19. The pathogenesis of COVID-19 includes an inflammatory storm with the release of cytokines from macrophages, which may be treated with anti-inflammatory drugs as celecoxib (CXB). For this, CXB-loaded PLGA microparticles (MPs) for inhaled therapy and that are able to be internalized by alveolar macrophages, were developed. MPs were prepared with 5% and 10% initial percentages of CXB (MP-C1 and MP-C2). For both systems, the mean particle size was around 5 µm, which was adequate for macrophage uptake, and the mean encapsulation efficiency was >89%. The in vitro release of CXB was prolonged for more than 40 and 70 days, respectively. The uptake of fluorescein-loaded PLGA MPs by the RAW 264.7 macrophage cell line was evidenced by flow cytometry, fluorescence microscopy and confocal microscopy. CXB-loaded PLGA MPs did not produce cytotoxicity at the concentrations assayed. The anti-inflammatory activity of CXB (encapsulated and in solution) was evaluated by determining the IL-1, IL-6 and TNF-α levels at 24 h and 72 h in RAW 264.7 macrophages, resulting in a higher degree of reduction in the expression of inflammatory mediators for CXB in solution. A potent degree of gene expression reduction was obtained with the developed CXB-loaded MPs.

## 1. Introduction

The current coronavirus pandemic, also known as COVID-19, is a severe acute respiratory infection caused by coronavirus-2 (SARS-CoV-2). It is highly transmissible, threatening human health and public safety. The most frequent initial signs are fever and respiratory distress with cough, followed in some cases, by dyspnoea, with infiltrates into the lungs [1,2,3].

Different studies have demonstrated that the pathogenesis of COVID-19 includes an inflammatory cascade of events, leading to the release of cytokines such as IL-1, IL-6, IL-12 and TNF-α, with similar characteristics to bacterial sepsis and hemophagocytic lymphohistiocytosis [4,5,6,7]. These cytokines may produce a dysfunction in the alveolar-capillary exchange that could result in pulmonary fibrosis and organ failure [8,9]. Thus, increased levels of pro-inflammatory cytokines and of other proteins such as LDH, ferritin, or C-reactive protein (CRP) have been associated with patients who require more intensive and prolonged treatments, including not only the use of therapeutic agents, but also oxygen therapy and mechanical ventilation [10].

Pharmacological approaches are aimed at decreasing the levels of inflammatory mediators and the cytokine storm, in order to prevent systemic and respiratory failures [1]. These include the use of corticosteroids, bronchodilators, cytokine antibodies, antiviral drugs, chloroquine and NSAIDs given by various systemic routes of administration and/or inhalation [11,12,13,14,15].

The lungs are one of the organs most frequently affected by COVID-19 infection, as target sites for inhaled treatments [16]. However, the inhaled therapies have some limitations, such as rapid pulmonary clearance that mainly affects drugs with short elimination half-lives, thereby requiring more frequent dosing intervals than in other administration routes. For this, there is a growing interest in prolonging the duration of action for drugs that are intended for inhalation therapy. In this regard, one of the approaches may be the use of microparticulate systems that are capable of improving the arrival of drugs to the deep lungs, and prolonging their release from the systems, thereby allowing for a reduction in the dosage frequency and/or the dose that in turn will result in increased therapeutic efficacy and reduced side effects [17,18]. 

Among the NSAIDs, celecoxib (CXB) is a selective COX-2 inhibitor that also exhibits anti-inflammatory activity based on the control of pro-inflammatory cytokines such as IL-1, IL-6 and TNF-α. For this, it is extensively used in acute and chronic inflammatory processes [8,9,19,20]. Moreover, different studies have demonstrated that CXB reduces prostaglandin E2 (PGE2) signals while stimulating recovery from normal to severe COVID-19 infection [14,21]. Due to these effects, CXB is considered to be a potential treatment for COVID-19 [13,14,21]. 

Although CXB is usually well tolerated, there are several side effects that must be taken into consideration, such as headache, somnolence, nausea, abdominal discomfort and heartburn [20,22]. Some studies also indicate that CXB can increase the incidence of major cardiovascular events such as myocardial infarction, the worsening of heart failure and cerebral thrombosis, which could be related to the dose and duration of therapy [23]. 

Alveolar macrophages reside in pulmonary alveoli and intervene in clearing the airways, detecting pathogens and pollutants, and regulating inflammatory processes such as COVID-19 [24,25]. Therefore, the targeting of CXB to alveolar macrophages would prevent its systemic biodistribution, enhancing efficacy and reducing adverse side effects. 

To improve the uptake of drugs by macrophages, the use of biodegradable polymeric microparticles (MPs) is an interesting approach. In this regard, MPs prepared with PLGA (poly (D.L-lactic-co-glycolic acid) are of interest due to the biodegradability and the biocompatibility characteristics of this polymeric material [26,27,28]. PLGA is approved for drug delivery purposes by the FDA and EMA regulatory agencies. 

The uptake of MPs by macrophages depends on several physicochemical characteristics such as composition, size, shape, surface chemistry, elasticity and stiffness [27,29,30]. For instance, the optimum particle diameter for this purpose ranges between 1 µm and 10 µm, as particles are able to activate phagocytosis by macrophages in order to eliminate foreign bodies from the organism [26,31,32,33]. The surface charge of particles is important for both the stability and interaction of particles with phagocytic cells. The coating of the particle surface with hydrophobic materials facilitates recognition by alveolar macrophages [28]. Particle shape influences the interaction with cells, starting from attachment to transport, and then to intracellular trafficking. In addition, particles with a mass density of around 1 g/cm^3^ and a size of between 1 and 3 µm have been used for deposition in lung airways [27].

The aim of this work was to design, develop and characterize CXB-loaded PLGA MPs destined for pulmonary administration and that are capable of being internalised by alveolar macrophages. By doing so, the anti-inflammatory efficacy of CXB at the site of action will be increased, reducing the cytokine cascade that is generated inside the alveolar macrophages. 

## 2. Materials and Methods

### 2.1. Materials

Celecoxib (CXB) was purchased from Hangzhou Onion Chemical Co., Ltd. (Hangzhou, China). PLGA Resomer^®^ RG 502 with a ratio of 50:50 poly(D.L-lactic-co-glycolic acid) (Mw 12,000 Da) was obtained from Evonik Industries AG (Essen, Germany). Dichloromethane (CH_2_Cl_2_) was acquired from Thermo Fisher Scientific Inc. (Madrid, Spain). Polyvinyl alcohol (PVA, Mw 30,000–70,000 Da) and fluorescein-5-isothiocyanate (FITC) were provided by Sigma-Aldrich Química, S.A. (Madrid, Spain). Mouse interleukin TNF-a, IL-1 and IL-6 ELISA kits were supplied by Cusabio Technology LCC (Houston, TX, USA). RevertAid H Minus First Strand cDNA Synthesis kit was obtained from Thermo Fisher Scientific (Madrid, Spain), and GeneAmp^®^ PCR System 9700 thermocycler was acquired from Applied Biosystems Hispania S.A. (Madrid, Spain). All solutions and buffers were prepared with deionized and distilled water (Q-POD^®^ Milli-Q system, Millipore, Madrid, Spain).

### 2.2. Development of CXB-Loaded PLGA Microparticles

MPs were prepared using the solvent extraction–evaporation technique from an O/W emulsion. For this, 200 mg of polymer PLGA 502 were dissolved in 4 mL of CH_2_Cl_2_, and then the corresponding amounts of CXB were added to obtain drug:polymer ratios of 5% (formulation MP-C1) and 10% (formulation MP-C2) (Table 1). The organic phase was added dropwise to 20 mL of the aqueous phase (0.5% PVA) using a Kinematica Polytron™ PT 10/35 GT homogenizer (Thermo Fisher Scientific, Madrid, Spain) at 8500 rpm for 5 min. The MP suspension was then stirred for 3 h at room temperature to allow for the evaporation of the organic solvent. The MPs were then vacuum filtered, washed with deionized water and freeze-dried for a period of 24 h (Lyo Quest^®^, Telsta Technologies S.L., Madrid, Spain). The freeze-drying conditions were 30 °C and 100 mTorr. Blank MPs (formulation MP-0) and fluorescent MPs (formulation MP-F loaded with FITC) were also prepared using the same procedure (Table 1). 

The formulations MP-0, MP-C1 and MP-C2 were used to evaluate the anti-inflammatory activity. The formulation MP-F was used for the phagocytic studies conducted in the macrophages. All formulations were prepared in triplicate.

### 2.3. Characterization of Microparticles

#### 2.3.1. Morphology and Size Distribution

All MP formulations were analysed for size distribution and morphology. The mean diameter and size distribution were determined via laser diffraction in a Microtrac^®^ 3500 system (Microtrac Inc., Montgomeryville, PA, USA). Particle size distributions were represented by volume distribution profiles. Morphological characterization of the particles was performed via scanning electron microscopy (SEM) at 20 KV (JEM 6335F microscopy. JEOL Ltd., Tokyo, Japan).

#### 2.3.2. Encapsulation Efficiency and Drug Loading

High-performance liquid chromatography (HPLC) was used for quantifying the amount of CXB encapsulated within the MPs. For this procedure, an exact amount (10 mg) of MPs was dissolved into 1 mL of dichloromethane (DCM). Ethanol (14 mL) was used to precipitate the polymer. The suspension formed was centrifuged at 13,000 rpm for 5 min, and the supernatant was removed and filtered through 0.45 μm filters. 

An HPLC 200 Perkin Elmer chromatographic system (Perkin Elmer, Waltham, MA, USA) equipped with the Emplower CromaNec XP v.1.0.4 software and a 235C diode array detector (Perkin Elmer, Waltham, MA, USA) was used. The mobile phase was prepared with methanol:water (75:25, *v/v*) filtered before use through 0.45 µm nylon filters, and degassed. A Gemini^®^ 5µm NX-C18 column (250 mm × 4.6 mm, 5 µm, Phenomenex, Alcobendas, Spain) was used for analysis. The flow rate was set at 1 mL/min, the injection volume was 50 µL, and the detection wavelength was 250 nm. The retention time of CXB was 11 min. The HPLC method showed no interferences between CXB and PLGA at the selected wavelength. All analyses were conducted at 25 ± 0.5 °C. The method was linear within the concentration range of 2.5–25.0 μg/mL. The limit of detection (LOD) was 0.45 μg/mL and the limit of quantification (LOQ) was 1.36 μg/mL All analyses were carried out in triplicate. 

The quantification of FITC encapsulated within FITC-loaded-PLGA MPs was performed using direct spectrophotometry at 488 nm. The method was linear within the concentration range of 1–10 μg/mL, with an LOD of 0.13 µg/mL and an LOQ of 0.38 µg/mL. 

Encapsulation efficiency (EE%) and drug loading (DL) of CXB within the MPs (formulations MP-C1 and MP-C2) were determined via HPLC, and via spectrophotometry for FITC (formulation MP-F). EE (%) was expressed as follows:EE% = amount of CXB or FITC encapsulated within MPs (mg) × 100/initial amount of CXB or FITC used in the preparation of MPs (mg).

DL was calculated as follows:DL = amount of encapsulated CXB or FITC (mg)/100 mg MPs

#### 2.3.3. In Vitro Release Studies

In vitro release of CXB from the formulations developed was studied in a Memmert WNB 45 water bath (Memmert GmbH Co., KG, Schwabach, Germany) at 37 ± 1 °C and constant agitation (100 rpm). For this, 20 mg of CXB-loaded PLGA MPs were suspended in 4 mL acetate buffer at pH 5 prepared with 3% sulphate lauryl sodium (SLS). This pH was selected to simulate the macrophage intraphagosomal pH [34]. At pre-determined times, the MP suspension was centrifuged at 3000 rpm for 5 min, with the supernatant removed and filtered through 0.45 μm filters and replaced with fresh medium. CXB was quantified via spectrophotometry at 252 nm or via HPLC when a higher sensitivity was required. The spectrophotometric method was linear within a range of 2.5–25.0 µg/mL, with a limit of detection (LOD) of 0.3 µg/mL and a limit of quantification (LOQ) of 0.87 µg/mL. Release tests were performed for 47 days and 75 days for the formulations MP-C1 and MP-C2, respectively. Tests were performed in triplicate. 

In addition, in vitro release studies of CXB from CXB-loaded PLGA MPs were performed using phosphate-buffered saline (PBS) (pH 7.4) containing 3% SLS, in order to determine the potential amount of CXB released at this physiological pH.

#### 2.3.4. Dry Powder Studies for Inhalation

The test known as an aerodynamic assessment of fine particles was employed to evaluate the particle characteristics of aerosol clouds that are produced by inhaled preparations. To conduct this test, apparatus A of the European Pharmacopoeia [35] was used. This glass impinger consists of two chambers; an upper chamber (stage 1), which is a 100 mL round bottom modified flask connected to a lower chamber (stage 2), consisting of a 250 mL conical flask that collects the dose capable of penetrating into the lungs. Aerodynamic assessments of the formulations MP-C1 and MP-C2 were conducted. Briefly, 7 mL and 30 mL of the solvent were introduced into the upper and lower impingement chambers, respectively. To dissolve the CXB-loaded PLGA MPs, 1 mL of DCM was placed in both chambers, and the volume was completed with 6 mL or 29 mL of ethanol to extract all of the CXB from the MPs (final volumes of 7 mL or 30 mL in the upper and lower chambers, respectively). The amount of CXB assayed was 100 mg, with the airflow adjusted to 60 ± 5 L/min. HPLC was used to quantify the amounts of CXB recovered from each chamber. This test was conducted in triplicate.

### 2.4. Studies in Cell Macrophages

Cell culture studies were performed in the RAW 264.7 mouse macrophage cell line (ATCC^®^ TIB-71TM, American Type Culture Collection, Manassas, VA, USA). The incubation conditions were 37 °C and a 5% CO_2_ humid environment in RPMI 1640 medium (Lonza^®^, Walkersville, MD, USA) supplemented with 10% bovine foetal serum (BFS) and 1% gentamicin. Cells were maintained in culture for 48 h. Once the cells reached 80–100% confluence they were centrifuged, and the supernatant was removed and adjusted to the desired concentrations. Cell viability, the uptake of MPs by macrophages and the effect of MPs in the anti-inflammatory response were evaluated.

#### 2.4.1. Cell Viability

This study was conducted to compare the cell viability of CXB-loaded PLGA MPs (formulations MPs-C1 and MPs-C2) with that of CXB in solution. The concentrations of the CXB solutions assayed were 12 µg/mL (CXB-S1), 24 µg/mL (CXB-S2) and 48 µg/mL (CXB-S3). The MPs were tested at a concentration of 1 mg/mL. This concentration was established from to the amounts of CXB released at 72 h from the formulations MP-C1 and MP-C2. At this time, the amount of CXB released from 2 mg of MPs was close to 24 µg, which corresponds to the amount used for the preparation of the solution CXB-S1. Moreover, two-fold (CXB-S2) and four-fold (CXB-S3) concentrations of CXB-S1 in solution were assayed. Blank PLGA MPs (MP-0) at a concentration of 1 mg/mL were also assayed. All formulations (solutions and MPs) were incorporated into 2 mL of RPMI 1640 medium.

The MTT assay (3-[4,5-dimethylthiazol-2-yl]-2,5 diphenyl tetrazolium bromide) was conducted in the RAW 264.7 mouse macrophage cell line. For this, 300,000 cells were placed in MW6-plates supplemented with RPMI medium, and incubated for 24 h. Then, the RPMI medium was removed and the cells were treated with CXB-loaded PLGA MPs (MP-C1 and MP-C2), blank PLGA MPs (MP-0), and CXB solutions (CXB-S1, CXB-S2 and CXB-S3). After 72 h, the cell macrophages were centrifuged at 1000 rpm for 5 min and the supernatant was removed. The cells were then suspended in 0.1 mL of an MTT solution at a concentration of 0.5 mg/mL. In this assay, MTT is reduced to an insoluble dark-blue formazan crystal due to the active mitochondria of live cells, whereas the inactivated mitochondria of dead cells are not able to reduce MTT. Formazan crystals were dissolved in DMSO (0.1 mL). Then, the absorbances were measured at 595 nm in an iMark™ microplate absorbance reader (BIO-RAD Laboratory S.A., Madrid, Spain). Untreated cells (Mock, value 1) were used as a reference to calculate the cell survival rates in the treated macrophages. Tests were conducted in triplicate.

#### 2.4.2. Phagocytosis Studies of Microparticles

##### Phagocytosis via Flow Cytometry 

Flow cytometry (Flow cytometer FACS Calibur, Becton Dickinson, San Jose, CA, USA) was the technique used to evaluate phagocytosis. Cell macrophages were placed in wells containing 3 × 10^6^ cells/well in RPMI 1640 medium. FITC-loaded PLGA MPs (formulation MP-F) were suspended in the medium at a concentration of 0.8 mg/mL, and incubated for 1, 5 and 24 h. At these pre-established times, the medium was removed and cells were washed with PBS and detached from the surface with the aid of trypsin. Finally, the cells were resuspended with PBS and propidium iodide. Cell viability and fluorescence intensity were determined using an FL1 probe (530/15 nm) at a 488 nm excitation wavelength. The experiments were conducted in triplicate.

##### Phagocytosis by Fluorescence Microscopy

The fluorescent formulation (MP-F) was incubated for 1, 5 and 24 h to study the uptake of MPs by macrophages. The macrophage cell line RAW 264.7 was prepared at a concentration of 2 × 10^6^ cells per MW6 well. A round coverslip was placed on the bottom of each well to allow for the adhesion of macrophages for a period of 24 h. At 1, 5 and 24 h, the medium was removed and replaced with formulation MP-F (concentration 0.8 mg/mL) suspended in fresh RPMI. At these times, the macrophages were washed with 1 mL PBS at pH 7.4 and fixed with 1 mL methanol at −20 °C for 15 min. The macrophages were kept in 1 mL PBS (pH 7.4) at 4 ± 0.2 °C. Then, samples corresponding to each of the formulations assayed were prepared by adding one drop of DAPI (4′,6-diamidino-2-phenylindole dihydrochloride) mounting medium (Sigma Aldrich Química, Madrid, Spain). The samples were analysed using fluorescence microscopy in the UV range (Nikon with NIS elements, Nikon Instruments Inc., Melville, NY, USA).

##### Phagocytosis via Confocal Microscopy

Confocal microscopy analysis was also conducted for the phagocytosis studies of the formulation MP-F. For this, the same cell line and culture conditions previously indicated were used. Cells were placed in individual plates with 150,000 cells/well, and incubated with formulation MP-F at a concentration of 0.8 mg/mL in RPMI medium. The incubation times were 24 h and 72 h. At each time, the medium from each well was removed and the cells were washed twice with PBS. Then, each well was treated with Hoechst 33342 solution to achieve nuclear staining, as it binds the nucleic acids of DNA in both living and fixed cells. Finally, the Hoechst/DNA complex was excited at 350 nm and at 461 nm emission wavelength. A Leica SP5 confocal laser scanning microscope (Leica Microsystems, Wentzler, Germany) was used to visualize the MPs. All tests were conducted in triplicate.

### 2.5. Anti-Inflammatory Activity of CXB-Loaded PLG MPs

#### 2.5.1. Determination of Inflammatory Cytokines

The anti-inflammatory activity of CXB encapsulated within MPs (MP-C1 and MP-C2) was compared to that of CXB in solution, by quantifying the protein expression of cytokines IL-1, IL-6 and TNF-α via ELISA (Model 680 Microplate Reader, BIO-RAD Laboratory S.A., Madrid, Spain), using a RAW 264.7 mouse macrophage cell line at the time periods of 24 h and 72 h. Blank MPs (MP-0) were also analysed.

To conduct this study, a concentration of 2 × 10^6^ cells per T-25 flask was prepared for each formulation assayed. The volume of the RPMI medium was 7 mL per flask, supplemented with 10% foetal bovine serum (FBS), and 1% gentamicin at 37 °C and 5% CO_2_. Prior to the addition of each formulation, the cell cultures were treated with LPS (lipopolysaccharide, 0.7 μg, concentration 0.1 µg/mL) to reproduce an inflammatory process. Macrophages treated only with LPS were used as a control (mock). All formulations (MP-0, MP-C1, MP-C2 and CXB solution) were incubated for 72 h. To resemble physiological conditions, the RPMI medium was removed every 24 h. 

The amount of MPs tested was 7 mg for each formulation (MP-C1, MP-C2 and MP-0), as selected from the results obtained at 72 h in the in vitro release tests performed at pH 5 with the formulations MP-C1 and MP-C2. For both formulations, the amount of CXB released at 72 h was similar (close to 80 μg). For the preparation of the CXB solution, the amount of active ingredient used was 80 μg. Moreover, 3.5 mg of the formulation MP-C2 (MP-C2-L) was also assayed. In all cases, 7 mL of culture medium was used. 

Finally, to determine the concentration of the inflammatory mediators, cell cultures were centrifuged for 10 min at 5500 rpm. The supernatants were analysed at 24 h and at 72 h via ELISA (mouse TNF-α ELISA kit, mouse IL-1 ELISA kit and mouse IL-6 ELISA kit, Cusabio Technology LLC., Houston, TX, USA). Cell macrophages were reserved for the determination of gene expression. 

#### 2.5.2. Determination of Gene Expression 

Macrophages reserved for determining gene expression were resuspended in 1 mL Trizol^®^ Reagent (Ambion, Life Technologies, Carlsbad, CA, USA) following the manufacturer’s instructions for the isolation of RNA, at 72 h. A GE NanoVue spectrophotometer (GE Healthcare Life Sciences, Hatfield, UK) was used to measure the RNA concentrations. The RNA absorbances of Mock, MP-0, MP-C1, MP-C2, MP-C2-L and CXB solution were measured at 260 nm and 280 nm. In all cases, the A260/A280 ratios were close to 2. The values were in the range of 1.8–2.1, thereby indicating high RNA purity [36]. 

##### Reverse Transcription-Polymerase Chain Reaction (RT-PCR)

For the RT-PCR analysis (Reverse Transcription Polymerase Chain Reaction), the total RNA was reverse-transcribed to a single strand of cDNA. This was performed using a Thermo Scientific RevertAid H Minus First Strand cDNA Synthesis kit (Thermo Fisher Scientific, Madrid, Spain), according to the supplier’s instructions, and using 1 µg of RNA from each sample. RT-PCR analysis was conducted in a 50 μL total sample volume using 1× Phusion Flash High-Fidelity PCR Master Mix (Thermo Fisher Scientific, Madrid, Spain), 0.2 μM of the specific primers and 2 μL of cDNA with its corresponding primers, depending on the transcript to be amplified (IL-1, IL-6, TNF-α or GADPH; Table 2). 

For this reaction, the GeneAmp^®^ PCR System 9700 thermocycler (Applied Biosystems, Waltham, MA, USA) was used with the following amplification parameters: 3 min at 95 °C; 40 cycles of 1 min at 95 °C, 1 min at 55 °C and 2 min at 72 °C; and a final extension step of 3 min at 72 °C. The results were visualized via electrophoresis in 1% agarose gels containing 0.005% of ethidium bromide, as described by Green and Sambrook [37].

The amplified fragments were quantified via densitometry using ImageJ software [38]. Moreover, a constitutive gene from mouse GADPH (glyceraldehyde-3-phosphate dehydrogenase) was amplified via RT-PCR in order to establish that the RNA samples contained almost the same number of cells, and thus to standardize the RNA concentration for the comparative analyses performed between the in vitro models.

### 2.6. Statistical Analysis

Data analysis was performed using Statgraphics software version 18 × 64-bit (Statgraphics Technologies, Inc., The Plains, VA, USA). Data were expressed as mean ± standard deviation (SD) from three different experiments, and analysed using a one-way analysis of variance (ANOVA). Statistically significant differences were defined as *p* < 0.05.

## 3. Results and Discussion

Celecoxib (CXB) is a selective COX-2 inhibitor that is extensively used to treat acute and chronic inflammation [19,20]. Due to its anti-inflammatory effects, CXB could also be potentially used to treat COVID-19 infection [13,14,21]. 

In pulmonary inflammatory processes, pro-inflammatory cytokines such as IL-1, IL-6 and TNF-α are involved [8,9]. Moreover, the macrophages are stimulated, thereby being considered as potential targets for therapy-based approaches based on the use of controlled drug delivery systems such as MPs. For this, the optimal particle sizes to activate phagocytosis by macrophages lie in the range of 1–10 µm [31,32,39].

### 3.1. Characterization of Microparticles

In our work, CXB-loaded PLGA MPs were developed using two different initial percentages of the drug (5% and 10%), resulting in formulations MP-C1 and MP-C2, respectively (Table 1). The method of preparation was solvent extraction–evaporation from an O/W emulsion. This method was previously optimized by our research group to obtain adequate particle sizes for the MPs [28]. Additionally, blank MPs (MP-0) and fluorescent FITC-loaded PLGA MPs were prepared (MP-F) (Table 1). 

Table 1 summarizes the values of drug loading (DL) and encapsulation efficiency (EE%) obtained for the formulations MP-C1 and MP-C2. To reduce the amount of MPs delivered by inhalation, high values of drug loading and encapsulation efficiency are desirable. The mean EE values for MP-C1 and MP-C2 were 89.35 ± 4.94% and 95.12 ± 3.98%, respectively. These high values may be related to the lipophobic nature of CXB [40], which exhibits greater affinity for the polymeric matrix than for the aqueous phase of the O/W emulsion formed during the encapsulation process [41]. The mean EE was slightly lower for the formulation MP-C1; however, non-statistically significant differences were found between both MP formulations (*p* > 0.05), thereby indicating that the increase in the initial amount of CXB used in the preparation of the MPs did not lead to a decrease in the EE, probably due to its high affinity for the polymer matrix. The mean values of DL were 4.28 ± 0.23 mg and 8.65 ± 0.62 mg of CXB in 100 mg of the formulations MP-C1 and MP-C2, respectively. The DL was almost double for the formulation MP-C2, which is very interesting as it will potentially allow for the administration of a higher dose of CXB, but the same amount of MPs. 

The particle size of MPs is a crucial factor when developing a controlled release system that is intended for inhalation when targeting alveolar macrophages [28,39]. In general, particles sizes in a range of 1–5 μm are deposited in small-diameter bronchioles, whereas sizes of a range of 1–2 μm are appropriate for systemic absorption in the alveolar epithelium [42,43].

The uptake of micro- and nano-systems by macrophages is conditioned by particle size. If particles are too small, as in the case of nanoparticles (nm), they may not be recognized by the mononuclear phagocyte system (MPS) [31,44]. In this case, active targeting may be necessary, which can be achieved by attaching a ligand to the particle surface that is capable of binding to a macrophage receptor and then being captured by the cells. On the other hand, if the particles are too large, the macrophages do not have the ability to capture them. Therefore, particle sizes in a range of 1–10 μm are the most suitable when being attained for macrophage phagocytosis [32]. In our case, mean particle sizes of 4.73 ± 0.46 µm and 4.40 ± 0.57 µm were obtained for the formulations MP-C1 and MP-C2, respectively (Table 1), also resulting in narrow size distributions and low inter-batch variability. Regarding the particle sizes, non-statistically significant differences were found with respect to the blank MPs (MP-0) (*p* > 0.05). With these particle sizes, the objectives of developing MPs for inhalation can be met. 

Figure 1 shows scanning electron microscopy (SEM) images of CXB-loaded PLGA MPs (MP-C1 and MP-C2) and blank MPs (MP-0). The particle size distributions of all formulations are also shown. All formulations were spherical with smooth surfaces, without deformations or pores, and without the presence of CXB crystals on their surfaces (Figure 1a,b). Homogeneous size distributions were obtained for all formulations, with polidispersity indexes (PDI) that were lower than 0.4. PDI values that are lower than 0.5 indicate homogeneous populations. 

In vitro CXB release tests were conducted for MP-C1 and MP-C2 at two different pH values of the dissolution medium: 5 and 7.4. A pH of 5 was selected to resemble the acidic lysosomal environment of the macrophages. In addition, pH 7.4 was selected as it is the physiological pH at which the MPs accessing the deep lung will remain before being potentially phagocytosed by the macrophages. For both MP formulations, the release of CXB was prolonged for more than 40 days, with marked differences found between both pH values. At pH 5, a more rapid release of CXB was obtained than at pH 7.4 (Figure 2). 

Figure 2 shows the release profiles obtained for MP-C1 and MP-C2 at pH 5. For both formulations, a biphasic profile was observed. The initial burst release (48 h) was around 28% and 14% for MP-C1 and MP-C2, respectively, with the amounts of CXB released at this time being practically the same. This rapid release was followed by a slow release occurring between days 2 and 17. During this phase, the amount of CXB released from both formulations was the same. Thereafter, the formulation of MP-C2 resulted in a much faster release of the drug, probably due to the fact that during the first phase (days 2 to 17), the release is mainly governed by diffusion, whereas during the second phase (days 17 to 42), both diffusion and erosion mechanisms intervene [45]. For both formulations, more than 94% of the CXB content was released after 47 days.

At pH 7.4, a sustained release of CXB was maintained for more than 70 days (Figure 2). The initial burst release (48 h) was around 18% and 11% for the formulations MP-C1 and MP-C2, respectively, which was followed by a very slow release between days 2 and 11. The duration of this phase was shorter than at pH 5, which could be explained by the fact that the diffusion process of CXB was clearly influenced not only by the particle size, but also by the pH. During this phase, the amounts of CXB released did not show differences between both formulations. After 11 days, the MP-C2 formulation was released faster the active compound. After 75 days, more than 90% of the CXB was released from both formulations. 

In the development of multiparticulate systems intended for inhalation therapy, it is necessary to ensure that the particles generated in the aerosol cloud have adequate sizes and aerodynamic characteristics to access the deep lung and to reach the alveoli. The targets for these systems are the alveolar macrophages that reside in the pulmonary alveoli and the inter-alveolar septum. In this regard, the aerodynamic assessment of the fine particles test was conducted with the formulations MP-C1 and MP-C2. The percentages of particles reaching stage 2 (deep lung) were 18.2 ± 1.9% and 15.7 ± 2.2%, respectively. These values are similar to those reported by other authors for powder inhalers (<20%) [46]. 

### 3.2. Studies in Cell Macrophages

Once administered, the MPs must reach the macrophages present in the alveoli and/or the interalveolar septum, and then be phagocytosed. As systemic biodistribution is avoided, only a small amount of inhalable MPs may be needed in order to obtain a therapeutic response if the active agent is effectively directed to these cells. Thus, the alveolar macrophages are the target cells for this inhaled therapy based on MPs.

In our study, cell viability in the macrophages and the effect of the uptake of PLGA MPs by macrophages, as well as the anti-inflammatory activity of CXB encapsulated within PLGA MPs (MP-C1 and MP-C2), were evaluated. 

Different authors have stated that the optimal sizes to activate phagocytosis by the macrophages are around 1–10 µm [31,32,39]. RAW 264.7 macrophages are frequently used as in vitro models for alveolar macrophages studies [47,48]. In our research, this cell line was selected to evaluate cell viability and the uptake of the MPs by the macrophages. 

We have also evaluated the potential cytotoxic effect of blank PLGA MPs (MP-0), CXB-loaded PLGA MPs (MP-C1 and MP-C2) and three CXB solutions prepared at concentrations of 12, 24 and 48 µg/mL (CXB-S1, CXB-S2 and CXB-S3). Figure 3 shows the cell viability results obtained. It can be seen that macrophages treated with the MP formulations (MP-0, MP-C1 and MP-C2) and solutions (CXB-S1 and CXB-2) presented similar values to those of the untreated cells (PC cells). Only in the case of the solution CXB-S3 was a reduction in cell viability of around 40% was obtained with respect to the control cells. However, if compared with the therapeutic range of CXB, this concentration would be too high [49]. 

Alveolar macrophages and other phagocytic cells differ in their ability to capture particles. For instance, alveolar macrophages preferentially phagocytose particles with sizes of around 3–6 μm [32]. The clearance of drugs from the deep lung is mainly conducted via particle uptake by alveolar macrophages and elimination through mucociliary clearance [50]. In our case, the entrapment of MPs inside macrophages would also allow for a longer permanence of CXB at its target site. 

To study the uptake of PLGA MPs by macrophages, the formulation MP-F was prepared (FITC-loaded-PLGA MPs, Table 1), as it can be detected inside the cells using confocal microscopy. SEM images (Figure 1d) revealed that MPs from the formulation MP-F have spherical shapes and smooth surfaces, with a mean particle size of 4.01 ± 0.89 μm and a narrow size distribution. This mean particle size is similar to those of the formulations MP-C1 and MP-C2, thereby being adequate for macrophage uptake. For the formulation MP-F, the EE was 1.29 ± 0.21% and DL was 0.12 ± 0.02 mg FITC/100 mg MPs (Table 1). These low values result from the high degree of water solubility of the fluorescent marker (600 mg/mL at 25 °C), which hinders its incorporation inside the MPs. 

Studies on the uptake of MPs by macrophages were conducted in RAW 264.7 macrophages by flow cytometry, fluorescence microscopy and confocal microscopy. 

The mean cell viability as estimated via flow cytometry was 98% and 96% at 1 h and 5 h, respectively. However, after 24 h, the cell viability decreased to 50%. Figure 4 shows the intensity of the fluorescence profiles obtained for the formulation MP-F at 1, 5 and 24 h. At all times, the values obtained were higher than those of their corresponding controls. As macrophages exhibit autofluorescence [51], the differences found between the formulation MP-F and the control cells could be attributed to the uptake of MPs by the macrophages. Our results suggest that the formulation MP-F was rapidly captured, remaining inside the cells for at least 24 h.

The uptake of MP-F by macrophages was also analysed using fluorescence microscopy. For this, the formulation MP-F was incubated for 1, 5 and 24 h with the RAW 264.7 macrophages cell line, according to the protocol described in Section 2.4.2. Figure 5 shows the images obtained. It can be observed that, as indicated by other authors [52,53,54], the macrophages exhibited autofluorescence due to the presence of NAD(P)H, flavins, porphyrin and lipofuscin.

A previous study conducted by our research group [55] on the uptake of FITC-loaded PLGA MPs by macrophages found that the particles were captured within short periods of time (≤5 h), although an analysis was not conducted within a longer incubation time (24 h), as in the present study. Therefore, the results obtained in our studies indicate a rapid uptake of MPs by macrophages (1 h) that is maintained for at least 5 h (Figure 5). At these times, particles from the formulation MP-F were observed as dark spots with a fluorescent halo within the cell macrophages. These halos resulted from the high degree of water solubility of FITC, which led to its rapid diffusion outside of the MPs. Therefore, at 24 h, this loss of fluorescence resulted in a more intense halo inside the macrophages. The uptake of formulation MP-F by the macrophages was not analysed at 72 h, due to the marked reduction in cell viability obtained at this time. 

These results, together with previous studies conducted by our research team [28] have demonstrated that flow cytometry is not able to distinguish between MPs that are entrapped within macrophages, and those adsorbed onto their surfaces. For this, confocal microscopy was also used. Figure 6 shows microphotographs of the formulation MP-F obtained via confocal microscopy at 24 h and 72 h. In this technique, the Hoechst solution used is a reagent for the fluorescent staining of DNA in live or fixed cells. This agent has a high specificity for binding to double-stranded DNA, preferentially to the A–T pairs. When this union occurs, a blue fluorescence is emitted, allowing for the observation of the nuclei of the macrophages. In our case, due to the rapid release of FITC from the MPs, its fluorescence was not detected. However, MPs were observed inside the cells (macrophages), which supports the results obtained using fluorescence microscopy. 

### 3.3. Anti-Inflammatory Activity of CXB-Loaded PLGA MPs

Once it was established that MPs with adequate particle sizes could be captured by macrophages, the anti-inflammatory activity of CXB encapsulated within the MP formulations was evaluated. The anti-inflammatory activity of CXB was related to the selective inhibition of the cyclooxygenase isoenzyme COX-2. This isoenzyme, together with the COX-1 isoform and phospholipase A, synthesizes prostaglandins (PGs) from arachidonic acid. Furthermore, PGE2 activates the expression of pro-inflammatory cytokines such as interleukins IL-1 and IL-6, and tumour necrosis factor-α (TNF-α) [56]. 

Macrophage-mediated responses may lead to tissue damage promoting inflammatory responses by secreting pro-inflammatory cytokines such as TNF-α, IL-1 and IL-6, among others [57]. In our study, as indicated in Section 2.5.1, LPS was used to reproduce an inflammatory process in macrophages as the lipopolysaccharide activates the production of pro-inflammatory cytokines [58,59], which also participate in the progression of COVID-19 infection [60]. 

The anti-inflammatory activity of CXB encapsulated within MPs (MP-C1 and MP-C2) was compared to those of CXB in solution and blank MPs (MP-0) by quantifying the expression of IL-1, IL-6 and TNF-α via ELISA using the RAW 264.7 mouse macrophage cell line at two incubation times (24 h and 72 h). Macrophages only treated with LPS were used as the control (Mock). 

As the anti-inflammatory activity was evaluated at 24 h and 72 h, the quantities of the formulations MP-C1 and MP-C2 assayed were selected from the amounts of CXB released at 72 h in the in vitro tests conducted at pH 5. At this time, the release profiles obtained for both formulations were practically similar. In addition, a lower dose (50%) of MP-C2 (formulation MP-C2-L) was also assayed to evaluate whether drug loading (DL) could have an influence on the anti-inflammatory response, as the DL of the formulation MP-C2 was almost twice that of MP-C1. 

For the CXB solution, the amount of CXB incorporated into 7 mL of culture medium was 80 μg, which is equivalent to the mean amount of CXB that would be released at pH 5 after 72 h (78.4 ± 2 μg) from 7 mg of MPs (formulation MP-C1 or MP-C2). The amount of the low-dose formulation MP-C2 (MP-C2-L) assayed was 3.5 mg. 

Figure 7 shows the mean concentrations (±SD) of IL-1, IL-6 and TNF-α obtained for all of the formulations assayed. As can be seen, the incorporation of LPS stimulated the production of all inflammatory mediators, resulting in high concentrations. Figure 7a shows the mean concentrations (±SD) of IL-1 obtained after exposing macrophages to all formulations. After 72 h, a slight reduction in concentration occurred for the blank MPs (MP-0). When the CXB formulations (MPs or solution) were assayed, a marked reduction in IL-1 levels was found with respect to the control cells (Mock) and blank MPs (*p* < 0.05). At this time, a slight increase in IL-1 levels was observed when compared with an incubation time of 24 h for all CXB-loaded MPs (MP-C1, MP-C2 and MP-C2-L), thereby suggesting a possible modification of the inflammatory response as the incubation time increases. 

The reduction in IL-1 levels was more marked for the formulation MP-C2 with respect to MP-C1 (*p* < 0.05), when the same amounts (7 mg) of each formulation were assayed. However, at the end of the study, the results obtained for the formulation MP-C2-L (3.5 mg) were similar to those corresponding to formulation MP-C1 (7 mg). The best results corresponded to formulation MP-C2, for which the reduction in IL-1 levels was higher than 90%.

When analysing the IL-6 concentrations, at both times (24 h and 72 h), the incubation of macrophages with blank MPs (MP-0) did not result in statistically significant differences with respect to the Mock cells (*p* > 0.05) (Figure 7b). At both times, the levels of IL-6 were significantly reduced by all of the CXB-containing formulations (MP-C1, MP-C2, MP-C2-L and the CXB solution) (*p* < 0.05), with this reduction being clearly more marked when CXB was encapsulated within MPs, in comparison to CXB in solution (*p* < 0.05). When the same amounts of the formulations MP-C1 and MP-C2 were compared (7 mg), the latter led to a greater reduction in IL-6 levels, with statistically significant differences between them (*p* < 0.05). However, when low-dose CXB-loaded MPs (MP-C2-L, 3.5 mg) were assayed, the results obtained were similar to those of formulation MP-C1 (7 mg), with a more than 88% reduction in IL-6 levels achieved by both formulations with respect to the Mock cells.

On the contrary, the TNF-α levels were higher with blank MPs (MP-0) at both times (Figure 7c). This result is in agreement with other studies [39,61,62], in which the presence of PLGA MPs led to a slight increase in this factor. The incorporation to the culture medium of CXB either encapsulated or in solution resulted in statistically significant decreases in TNF-α levels (*p* < 0.05), which were especially marked when CXB was encapsulated (MP-C1, MP-C2 and MP-C2-L), and for which mean decreases of 81.6%, 93.5% and 86.7%, respectively, were obtained with respect to the blank MPs (MP-0). 

It can therefore be stated that all CXB-loaded MPs significantly decreased the levels of the pro-inflammatory mediators analysed (IL-1, IL-2 and TNF-α), also achieving better results than CXB in solution. Therefore, a more potent anti-inflammatory response was obtained, especially with the formulation MP-C2, which was prepared with an initial ratio of 1:10 CXB:PLGA. These results could be attributed to either the faster in vivo release of CXB within macrophages than that observed in the in vitro tests, or because the targeting of CXB to macrophages is more efficient when given encapsulated within MPs than in solution.

Finally, Figure 8 shows the results obtained from the RT-PCR analysis conducted. The mock band was chosen as a control, given a value of 1, and used as a reference to calculate the intensity values of the other bands. A negative control (NC, value 0) was also used to assure that both the materials and the methodology did not have any influence in the results obtained.

Blank MPs (MP-0) slightly decreased IL-1 gene expression when compared to the Mock band. All CXB-containing formulations markedly decreased in IL-1 gene expression, with the best results being obtained with the formulation MP-C2 (Figure 8a). The resulting decreases were 80%, 90%, 80% and 70% for the formulations MP-C1, MP-C2, MP-C2-L and CXB solution, respectively.

The blank MPs (MP-0) did not modify IL-6 gene expression, but all of the formulations prepared with CXB (MP-C1, MP-C2, MP-C2-L and CXB solution) led to marked inhibitions of IL-6 gene expression, which were complete inhibitions for all CXB-loaded MPs, and 99% inhibition when the CXB solution was tested (Figure 8b). As can be seen in Figure 8c, the same behaviour was observed for the gene expression of TNF-α, which was completely inhibited by all CXB-containing formulations.

## 4. Conclusions

Inhalation therapy is gaining increasing attention for the delivery of drugs destined to treat respiratory disorders associated with cytokine storms, such as COVID-19 infection. CXB has proven to be an interesting drug for the control of this cytokine storm in lung diseases. The design of a new controlled release system that is capable of targeting CXB to alveolar macrophages when given by inhalation could be an interesting alternative, as lower doses may be administered, resulting in equivalent or even superior efficacy than higher doses given by other routes. In addition, inhalation therapy can result in a more rapid onset of effect in the lung, and reduced side effects. The controlled release system developed (CXB-loaded PLGA MPs) produced a marked decrease in the levels of IL-1, IL-6 and TNF-α when compared to CXB in solution. These results could be attributed to either the faster in vivo release of CXB within macrophages than that observed in the in vitro tests, or because the targeting of CXB to macrophages is more efficient when given encapsulated within MPs than in solution. The results obtained with the new controlled release system of CXB are promising regarding its potential development and evaluation in further advanced phases of experimentation.

## Figures and Tables

**Figure 1 pharmaceutics-14-01392-f001:**
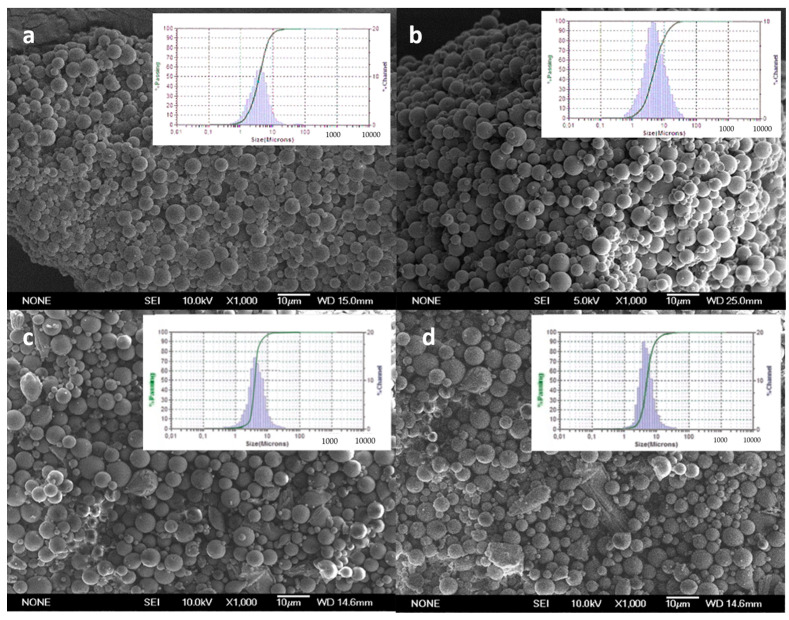
SEM microphotographs and particle size distributions of formulations. (**a**) MP-C1, (**b**) MP-C2, (**c**) MP-0 and (**d**) MP-F.

**Figure 2 pharmaceutics-14-01392-f002:**
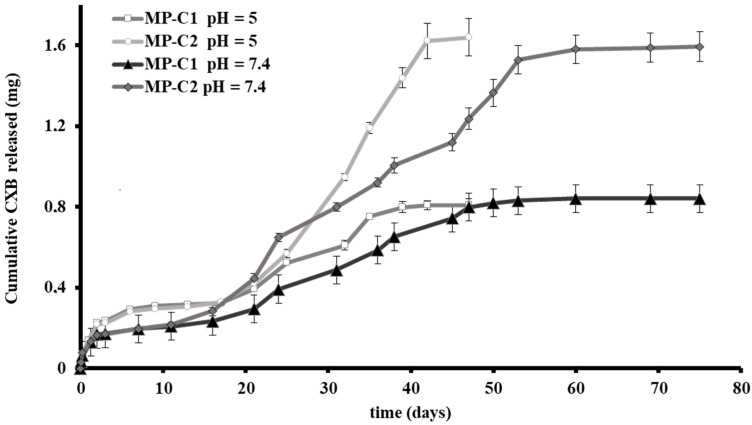
Cumulative amounts (±SD) of CXB released from CXB-loaded PLGA MPs (MP-C1 and MP-C2) at pH 5 and pH 7.4.

**Figure 3 pharmaceutics-14-01392-f003:**
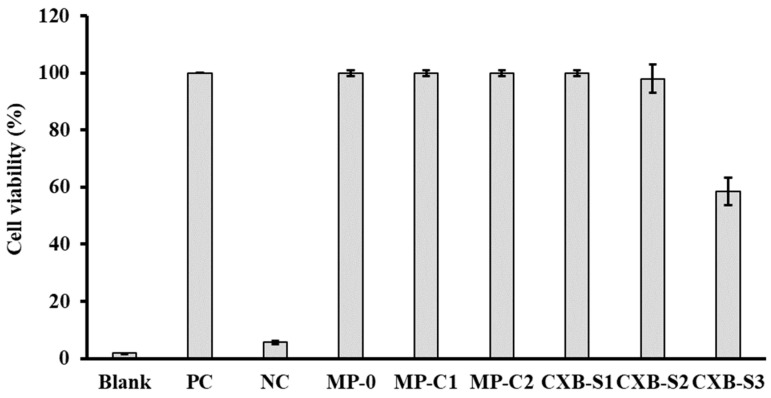
Cell viability results (±SD) obtained after the incubation of macrophages with formulations MP-0, MP-C1, MP-C2, CXB-S1 (12 µg/mL), CXB-S2 (24 µg/mL) and CXB-S3 (48 µg/mL). PC: positive control (live macrophages), NC: negative control (dead macrophages).

**Figure 4 pharmaceutics-14-01392-f004:**
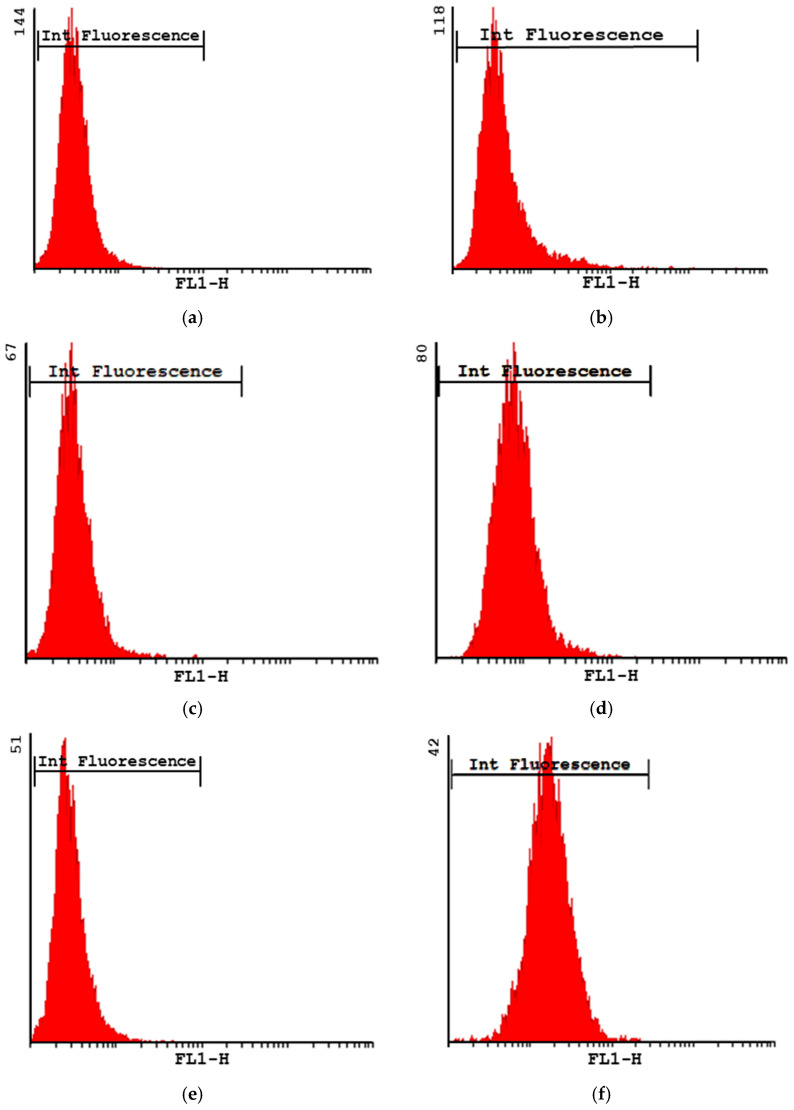
Intensity of fluorescence plots obtained for formulation MP-F at 1 h (**b**), 5 h (**d**) and 24 h (**f**), and their respective controls at 1 h (**a**), 5 h (**c**) and 24 h (**e**). MP-F: FITC-loaded PLGA MPs. FITC: fluorescein-5-isothiocyanate.

**Figure 5 pharmaceutics-14-01392-f005:**
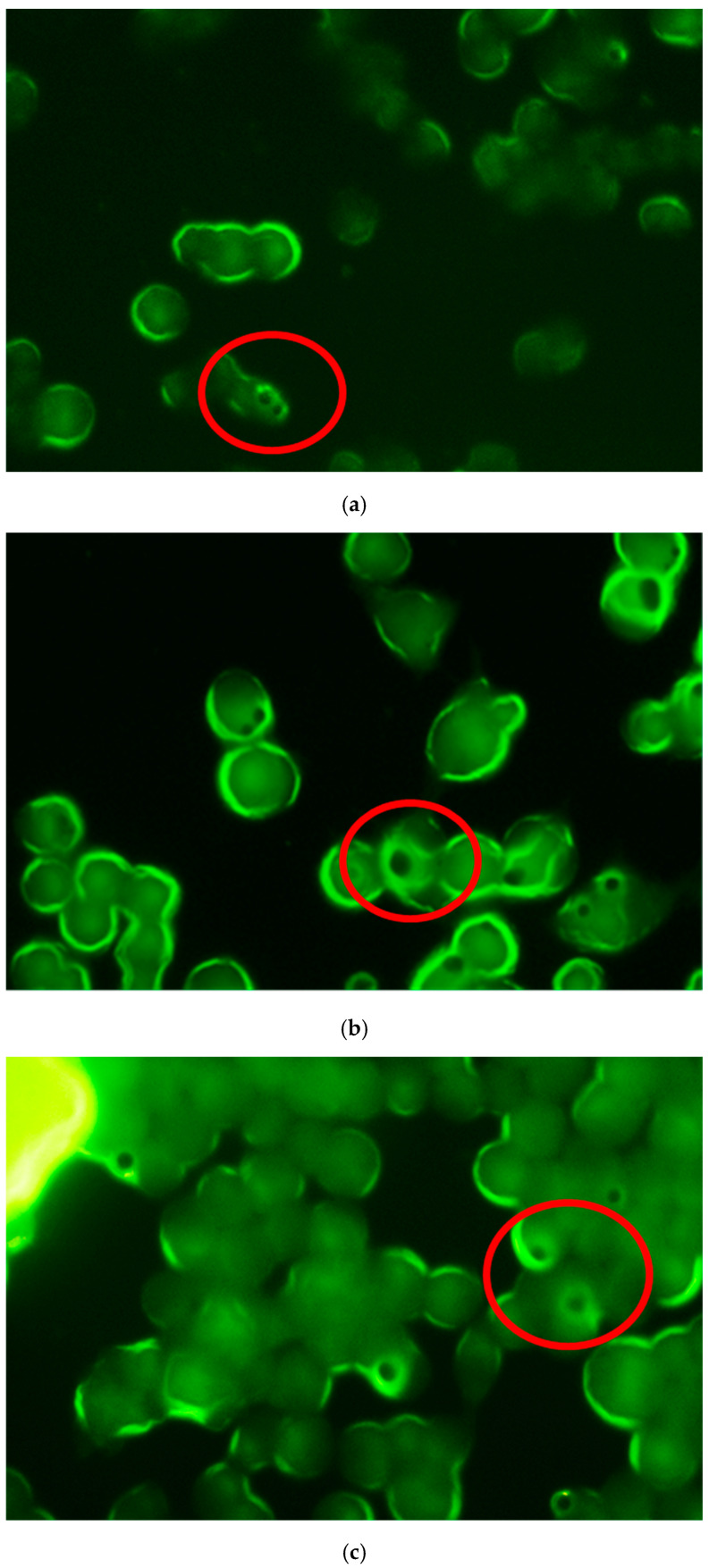
Fluorescence microscopy images of RAW 264.7 macrophages acquired at different times after incubation with formulation (**a**) MP-F: 1 h, (**b**) 5 h and (**c**) 24 h. MP-F: FITC-loaded PLGA MPs. FITC: fluorescein-5-isothiocyanate.

**Figure 6 pharmaceutics-14-01392-f006:**
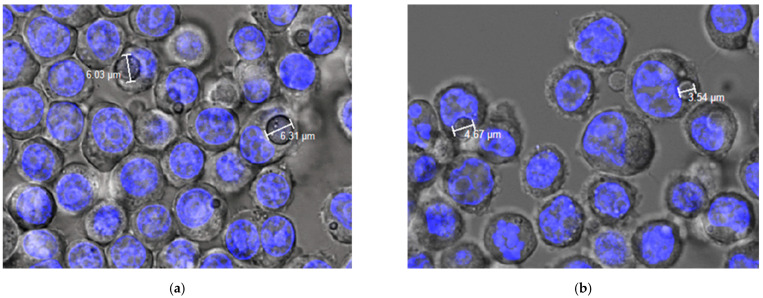
Confocal images of phagocytosis of formulation MP-F obtained at (**a**) 24 h and (**b**) 72 h. MP-F: FITC-loaded PLGA MPs. FITC: fluorescein-5-isothiocyanate.

**Figure 7 pharmaceutics-14-01392-f007:**
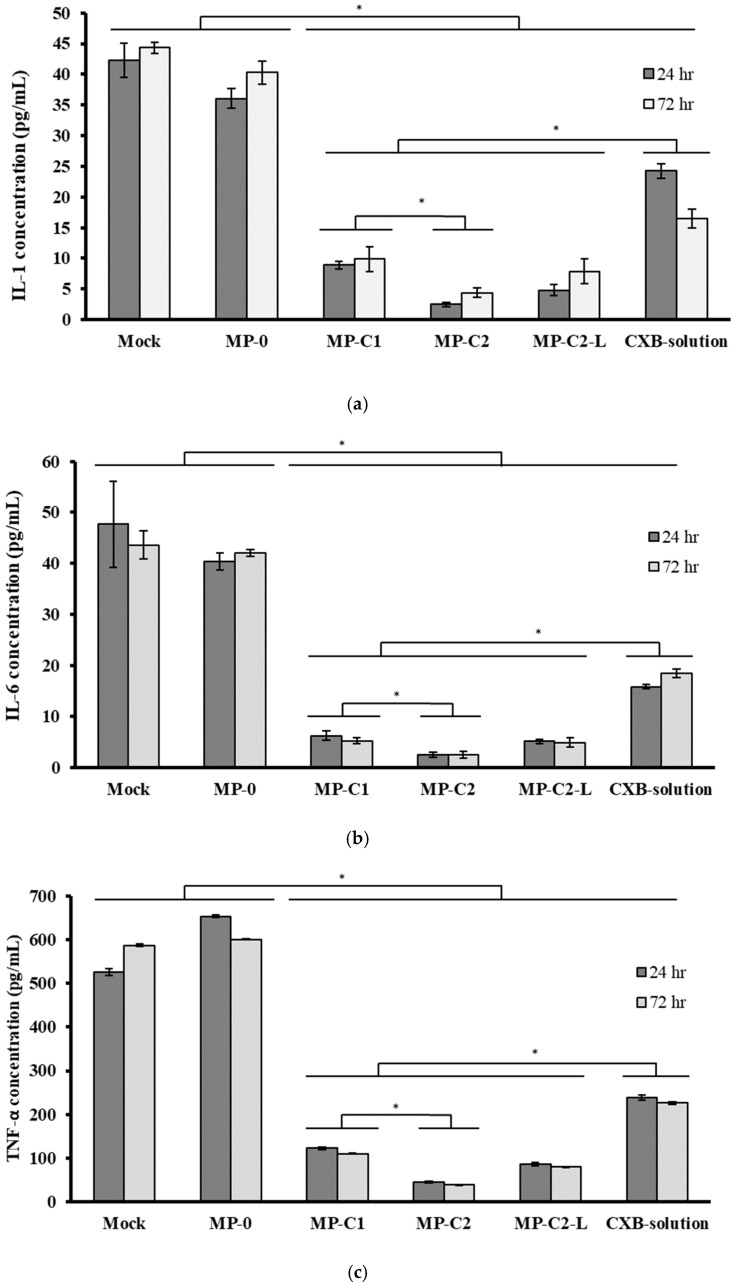
Mean concentrations (±SD) of (**a**) IL-1, (**b**) IL-6 and (**c**) TNF-α obtained after incubating macrophages with formulations MP-0, MP-C1, MP-C2, MP-C2-L and CXB solution for 24 h and 72 h. Mock: control cells. * *p* < 0.05.

**Figure 8 pharmaceutics-14-01392-f008:**
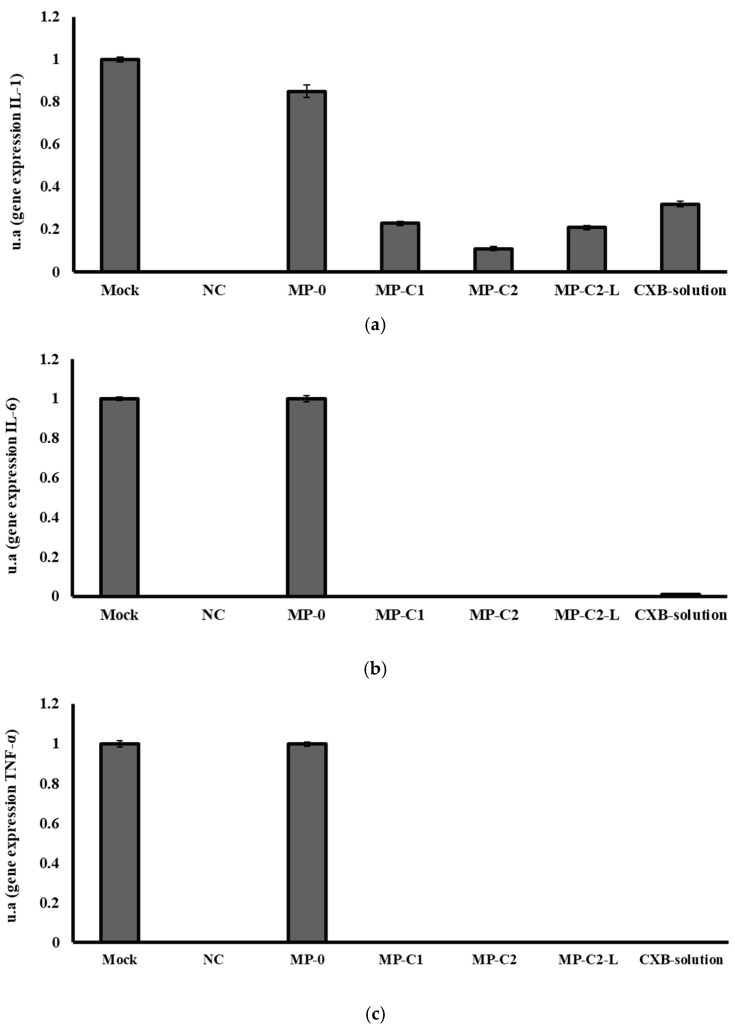
mRNA gene expression levels of (**a**) IL-1, (**b**) IL-6 and (**c**) TNF-α obtained after incubation for 72 h with formulations MP-0, MP-C1, MP-C2, MP-C2-L and CXB solution. Mock: control. NC: negative control.

**Table 1 pharmaceutics-14-01392-t001:** Composition of microparticle formulations developed, and their encapsulation efficiencies (EE%), drug loading (DL) and particle size. CXB (celecoxib) and FITC (fluorescein-5-isothiocyanate).

Formulation	CXB (%)	FITC (%)	Ratio Drug:Polymer	EE ± DS (%)	DL ± DS (mg of CXB in 100 mg of MPs)	Particle Size ± DS (μm)
MP-C1	5	-	1:20	89.35 ± 4.94	4.28 ± 0.23	4.73 ± 0.46
MP-C2	10	-	1:10	95.12 ± 3.98	8.65 ± 0.62	4.40 ± 0.57
MP-0	-	-	-	-	-	4.17 ± 0.45
MP-F	-	10	1:10	1.29 ± 0.21	0.12 ± 0.02	4.01 ± 0.89

**Table 2 pharmaceutics-14-01392-t002:** Forward and reverse primers of GADPH, IL-1, IL-6 and TNF-α that were used for RT-PCR.

	5′-Forward-3′	5′-Reverse-3′
IL-1	AGTTGACGGACCCCAAAAGAT	GTTGATGTGCTGCTGCGAGA
IL-6	CTTCCATCCAGTTGCCTTCTTG	AATTAAGCCTCCGACTTGTGAAG
TNF-α	GATCTCAAAGACAACCAACATGTG	CTCCAGCTGGAAGACTCCTCCCAG
GADPH	TGAGGCCGGTGCTGAGTATGTCG	CCACAGTCTTCTGGGTGGCAGTG

## Data Availability

Not applicable.

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
