# Peer review of "Celecoxib Microparticles for Inhalation in COVID-19-Related Acute Respiratory Distress Syndrome"

_pharmaceutics, 2022, doi:10.3390/pharmaceutics14071392_

Round 1
Reviewer 1 Report
In this paper, the authors was to design, develop and characterize CXB-loaded PLGA MPs destined for pulmonary administration. The results proved that the controlled release system
developed (CXB-loaded PLGA MPs) produced a marked decrease in the levels of IL-1, IL- 6 and TNF-α when compared to CXB in solution. And the results obtained with the new controlled release system of CXB are promising regarding its potential development and evaluation in
further advanced phases of experimentation. Although their results are interesting, there are a number of issues for the authors to consider for the improvement.
Majors:
1.In line 82, the authors mentioned that “The uptake of MPs by macrophages depends on several physicochemical characteristics such as composition, size, shape, surface chemistry, elasticity, and stiffness”, but in the following part, the author only mentions the influence of size on the uptake of MPs by macrophages, the other influences are not involved.
2.In “Development of CXB-loaded PLGA microparticles” part, the authors mentioned “The MPs were then vacuum filtered, washed with deionized water and freeze-dried for a period of 24 hr (Lyo Quest® , Telsta Technologies S.L., Madrid, Spain).” What is the condition for freeze-dried process, please add more details.
3.In “Encapsulation efficiency and drug loading” part, the determination method of encapsulation efficiency and drug loading is not clear and detailed enough. Please add more details.
4. In Results and Discussion part, I suggest the author divide this part into different parts consistent with materials and methods part.
5. In “In vitro CXB release tests” part, the authors did a comparison of MP-C1 and MP-C2 releases, but why no comparison with CXB solutions?
Minors:
1. Table 1 should remove to the Results and Discussion part.
2. In line 357, the authors mentioned that “In our case, mean particle sizes of 4.73 ± 0.46 µm and 4.40 ± 0.57 µm were obtained for formulations MP-C1 and MP-C2, respectively (table 1), also resulting in narrow size distributions and low inter-batches variability.”, What is the DPI of these formulations?
3. In line 380,the authors choose two different pH values of the dissolution medium: 5 and 7.4.
I don't understand why the authors did the experiment for 48 hours in pH 5 dissolution medium but did the experiment for 72 hours in pH 7.4 dissolution medium?
Author Response
Reviewer 1
Majors:
1. In line 82, the authors mentioned that “The uptake of MPs by macrophages depends on several physicochemical characteristics such as composition, size, shape, surface chemistry, elasticity, and stiffness”, but in the following part, the author only mentions the influence of size on the uptake of MPs by macrophages, the other influences are not involved.
Answer: The influence of several factors has been added to the text (see lines 86 to 91).
2. In “Development of CXB-loaded PLGA microparticles” part, the authors mentioned “The MPs were then vacuum filtered, washed with deionized water and freeze-dried for a period of 24 hr (Lyo Quest® , Telsta Technologies S.L., Madrid, Spain).” What is the condition for freeze-dried process, please add more details.
Answer: Freeze drying conditions have been added to the text (see line 120).
3. In “Encapsulation efficiency and drug loading” part, the determination method of encapsulation efficiency and drug loading is not clear and detailed enough. Please add more details.
Answer: estimation of the encapsulation efficiency and drug loading have been explained by the equations used for their calculations (see lines 160 to 164). Moreover, the retention time of CXB in the HPLC method has been included in the text (see line 149-150).
- In Results and Discussion part, I suggest the author divide this part into different parts consistent with materials and methods part.
Answer: The results and discussion section has been divided into different parts as indicated by the reviewer (see lines 337, 421 and 509).
- In “In vitro CXB release tests” part, the authors did a comparison of MP-C1 and MP-C2 releases, but why no comparison with CXB solutions?
Answer: Comparison with CXB in solution cannot be done as the drug is already dissolved in the solution at time 0.
Minors:
- Table 1 should remove to the Results and Discussion part.
Answer: Table 1 was included in the material and methods section of the manuscript in order to clarify and explain the composition of each MPs formulation developed. If moved to the results and discussion section, it will be difficult to understand the composition and differences among the MPs formulations.
- In line 357, the authors mentioned that “In our case, mean particle sizes of 4.73 ± 0.46 µm and 4.40 ± 0.57 µm were obtained for formulations MP-C1 and MP-C2, respectively (table 1), also resulting in narrow size distributions and low inter-batches variability.”, What is the DPI of these formulations?
Answer: The polydispersity index (PDI) has been indicated in the text (see lines 381-383).
- In line 380, the authors choose two different pH values of the dissolution medium: 5 and 7.4. I don't understand whythe authors did the experiment for 48 hours in pH 5 dissolution medium but did the experiment for 72 hours in pH 7.4 dissolution medium?
Answer: release tests were carried out for 48 hours in pH 5 and for 72 hours in pH 7.4 as at these times complete release of CXB was achieved. As it can be seen in figure 2 at these times both release profiles become asymptotic.
Reviewer 2 Report
The manuscript 1732384 entitled “Celecoxib microparticles for inhalation in COVID-19-related acute respiratory distress syndrome” reports the production of Celecoxib (CXB) microparticles prepared with PLGA (poly (D.L-lactic-co-glycolic acid) for potential use by inhalation against COVID-19. The microparticles were fully characterized, in vitro tests were performed using the RAW 264.7 mouse macrophage cell line 188 (ATCC® TIB-71TM). The anti-inflammatory activity of CXB MPs was examined by quantitation of the protein expression of cytokines IL-1, IL-6 and TNF-α . The expression of genes coding for IL-1, IL-6, TNF-α and GADPH was performed by RT-PCR. The results sound scientific and the conclusions are based in the observed data. The manuscript shows quality to be published in the present form.

Author Response
Reviewer 2
No comments.
Reviewer 3 Report
The article titled ‘’ Celecoxib microparticles for inhalation in COVID-19-related acute respiratory distress syndrome’’ by Monica-Carolina et al., is a good article. Authors have developed a novel formulation for the treatment of ARDS and explored it in vitro. Although it is a good study, the authors could have done in vivo studies in LPS-induced or oleic acid-induced ARDS animal model. Few more suggestions to improve the manuscript
1. Use of COVID-19 in the title looks misleading. ARDS can be because of a number of reasons. In COVID-19 there is cytokine storm that necessitates rapid availability of high level of anti-inflammatory medicine. Authors have made a sustained release product, which I Feel, is not suitable for COVID-19.
2. Please clarify whether you have used any surfactant while making spherical microspheres of PLGA?
3. Why Aerodynamic diameter was not measured? It is very essential for inhalation drug delivery. Use Adndersen Cascade Impactor. Refer few papers for more clarity. For example:
https://www.ncbi.nlm.nih.gov/pmc/articles/PMC8625129/
4. Why authors have not done studies in any Lung Cell Line?
5. Please mention the concentration of LPS used to induce inflammation?
6. Was FBS added to medium while performing inflammation studies?]
7. Try to make the paper precise. For example figure 2 graphs for 2 different pH conditions can be represented in 1 graph. It will be more clear for the comparison.
Author Response
Reviewer 3
The article titled ‘’ Celecoxib microparticles for inhalation in COVID-19-related acute respiratory distress syndrome’’ by Monica-Carolina et al., is a good article. Authors have developed a novel formulation for the treatment of ARDS and explored it in vitro. Although it is a good study, the authors could have done in vivo studies in LPS-induced or oleic acid-induced ARDS animal model. Few more suggestions to improve the manuscript
- Use of COVID-19 in the title looks misleading. ARDS can be because of a number of reasons. In COVID-19 there is cytokine storm that necessitates rapid availability of high level of anti-inflammatory medicine. Authors have made a sustained release product, which I Feel, is not suitable for COVID-19.
Answer: we agree with the reviewer however, in our case, burst release (first 48 hours) of CXB from the MPs together with their rapid phagocytosis by macrophages leads to a rapid reduction of the mRNA expression levels that is equivalent or even better to that obtained with the administration of CXB solution in an aerosol. Nevertheless, further studies should be conducted. In fact, we are already planning to perform further experiments in an ARDS animal model.
- Please clarify whether you have used any surfactant while making spherical microspheres of PLGA?
Answer: The surfactant used was 0.5% PVA (polyvinyl alcohol) as it was indicated in the text (see line 114). This surfactant was a key factor to obtain adequate particle sizes for our MPs.
- Why Aerodynamic diameter was not measured? It is very essential for inhalation drug delivery. Use Adndersen Cascade Impactor. Refer few papers for more clarity. For example: https://www.ncbi.nlm.nih.gov/pmc/articles/PMC8625129/
Answer: we agree with the reviewer however; we have used apparatus A of the European Pharmacopoeia as it is the only one available in our facilities.
- Why authors have not done studies in any Lung Cell Line?
Answer: In our work studies were not performed in lung cell lines. However, RAW 264.7 mouse macrophage cell line was used as target cells for our MPs formulations. Citotoxicity studies in lung cell lines would have been interesting however, we have previous experience with CXB in cytotoxicity studies performed in different cell lines in which at the same concentration ranges demonstrated to be non-cytotoxic.
- Please mention the concentration of LPS used to induce inflammation?
Answer: an error was detected as 0.7 µg of LPS were added instead of 0.7 µL. This error has been corrected in the text and the concentration indicated (see line 277).
- Was FBS added to medium while performing inflammation studies?
Answer: Yes, FBS was added to the medium. This has been included in the text (see lines 274 and 275).
- Try to make the paper precise. For example figure 2 graphs for 2 different pH conditions can be represented in 1 graph. It will be more clear for the comparison.
Answer: the figure has been modified according to the indications given by the reviewer.